# Casein-Hydrolysate-Loaded W/O Emulsion Preparation as the Primary Emulsion of Double Emulsions: Effects of Varied Phase Fractions, Emulsifier Types, and Concentrations

Pelin Salum [1], Çağla Ulubaş [1], Onur Güven [2], Levent Yurdaer Aydemir [1] and Zafer Erbay [1,*]

1   Department of Food Engineering, Faculty of Engineering, Adana Alparslan Turkes Science and Technology University, Adana 01250, Turkey
2   Department of Mining Engineering, Faculty of Engineering, Adana Alparslan Turkes Science and Technology University, Adana 01250, Turkey
*   Correspondence: zafererbay@yahoo.com; Tel.: +90-322-4550000 (ext. 2080)

**Abstract:** Stable primary emulsion formation in which different parameters such as viscosity and droplet size come into prominence for their characterization is a key factor in W/O/W emulsions. In this study, different emulsifiers (Crill™ 1, Crill™ 4, AMP, and PGPR) were studied to produce a casein-hydrolysate-loaded stable primary emulsion with lower viscosity and droplet size. Viscosity, electrical conductivity, particle size distribution, and emulsion stability were determined for three different dispersed phase ratios and three emulsifier concentrations. In 31 of the 36 examined emulsion systems, no electrical conductivity could be measured, indicating that appropriate emulsions were formed. While AMP-based emulsions showed non-Newtonian flow behaviors with high consistency coefficients, all PGPR-based emulsions and most of the Crill™-1- and -4-based ones were Newtonian fluids with relatively low viscosities (65.7–274.7 cP). The PGPR-based emulsions were stable for at least 5 days and had D(90) values lower than 2 μm, whereas Crill™-1- and -4-based emulsions had phase separation after 24 h and had minimum D(90) values of 6.8 μm. PGPR-based emulsions were found suitable and within PGPR-based emulsions, and the best formulation was determined by TOPSIS. Using 5% PGPR with a 25% dispersed phase ratio resulted in the highest relative closeness value. The results of this study showed that PGPR is a very effective emulsifier for stable casein-hydrolysate-loaded emulsion formations with low droplet size and viscosity.

**Keywords:** double emulsion; W/O emulsion; emulsifier; encapsulation

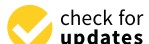



## 1. Introduction

In recent years, significant progress has been made in developing food-derived bioactive ingredients, and one important ingredient group of compounds in this context is bioactive peptides [1,2]. While bioactive peptides are inactive in protein structures, they can show a wide variety of physiological effects with their hormone-like properties when they are released from the protein structure by hydrolysis [3–5]. In general, these compounds contain 2–20 amino acid residues and show a variety of biological properties, depending on their structural properties, amino acid composition, sequence, and charge [6,7]. Milk proteins, especially caseins, are known to be a good source of bioactive peptides with anti-hypertensive, antidiabetic, antiobesity, antioxidant, immunomodulatory, mineral-binding, opioid, and antimicrobial properties [7–10].

However, there are several difficulties in the development of food products fortified with bioactive peptides [11]. The difficulties in the use of peptides arise from their low solubility, chemical and physical instability, undesirable flavor properties (especially bitter taste), and low bioavailability [12–14]. To overcome these problems, encapsulation technology presents promising solutions; however, encapsulation of bioactive peptides is a

challenging area with its specific features [15]. Among encapsulation methods, a technique with high potential for the encapsulation of bioactive peptides is the use of double emulsions [15–18].

Double emulsions are liquid dispersion systems in which the droplets of one emulsion's dispersed phase contain smaller dispersed droplets [19–21]. While the internal emulsion, generally termed in the production as "primary emulsion", can be described as the dispersed phase of the double emulsion, the whole emulsion system including the outer continuous liquid phase is generally termed as "secondary emulsion". They can be in two main morphologies such as oil-in-water-in-oil (O/W/O) or water-in-oil-in-water (W/O/W), and the latter one is used more commonly in the literature [21]. A two-stage emulsification technique is commonly used in the preparation of W/O/W emulsions. In the first stage, a stable primary emulsion (W/O) is obtained using lipophilic emulsifiers. It is important to obtain small and monodispersed water droplets in the oil phase. In the second step, the primary emulsion is dispersed in an external water-based continuous phase containing hydrophilic emulsifiers and stabilizers [22]. W/O/W systems have been used for the encapsulation of several types of protein hydrolysates and bioactive peptides with high efficiency [1,15–18,23–27]. However, the main problem in food-related applications of W/O/W emulsions is their long-term instability due to the limited variety of food-grade substances that can be used as emulsifiers or stabilizers [28]. On the other hand, the characteristics of primary emulsion play a significant role in the production of stable W/O/W [29]. Therefore, one of the approaches for improving the stability of double emulsions is developing a primary W/O emulsion with enhanced stability [19].

In this context, one reason for the low stability of W/O emulsions is the high mobility of water droplets. In W/O emulsions, only steric forces stabilize the emulsion due to the low electrical conductivity of the continuous phase [30]. The ratio of the dispersed phase, the type and concentration of the emulsifier, the properties of the oil, the presence of osmolyte, and the mechanism and processing conditions of homogenization affect the emulsion droplet size, viscosity, and thus stability [25,31,32]. Another important point for the stability of W/O/W emulsions is the properties of the dispersed water phase of the primary emulsion. The composition of this water phase and the presence of other compounds influence the final stability of the double emulsion [33]. For instance, it is well known that peptides exhibit interfacial activity and this interfacial activity can lead to unstable emulsions due to the interaction of the peptides with the lipophilic emulsifier at the water/oil interface [24]. Additionally, the W/O needs to be dispersed with a high internal droplet yield and smaller droplets in the external water phase [28]. To achieve this, a reduction in the viscosity of the primary emulsion is an option. The primary emulsion with low viscosity supports the formation of small W/O droplets in a double emulsion with a low-energy homogenization process, and the primary emulsion droplets can be easily dispersed in the continuous water phase [24,34].

The present work aimed to produce a stable peptide-loaded W/O emulsion with low droplet size and viscosity that can be used as a primary emulsion in W/O/W double emulsions for the encapsulation of casein hydrolysates. For this purpose, different emulsifiers were used in the preparation of W/O primary emulsions at various emulsifier concentrations and dispersed phase ratios.

## 2. Materials and Methods

### 2.1. Materials

The skimmed raw cow's milk was obtained from Sarıçam Ali Baba'nın Çiftliği Milk and Dairy Products Company in Adana, Turkey and used in the preparation of acid casein. Alcalase® 2.4 L was obtained as a gift sample kindly provided by Novozymes (Bagsvaerd, Denmark). Sunflower oil was purchased from a local store. Sorbitan monolaurate (Crill™ 1), and sorbitan monooleate (Crill™ 4) were kindly supplied by Croda Chemicals (Snaith, UK). Ammonium phosphatide (AMP 4455) and polyglycerol polyricinoleate (PGPR 4150) were obtained as gift samples kindly provided by Palsgaard® (Juelsminde, Denmark). The chem-

ical structure of the emulsifiers used in the present study was presented in Figure 1 [35–37] and the critical micelle concentrations of Crill™ 1, Crill™ 4, and PGPR were reported in the literature as $2.1 \times 10^{-5}$, $1.8 \times 10^{-5}$, and $9.0 \times 10^{-3}$ mol/L, respectively [38,39]. Acetic acid, trisodium citrate, trisodium phosphate, and potassium sorbate were purchased from Sigma-Aldrich (St Louis, MO, USA).

$R_1$: $C_{11}H_{23}$ (for Crill 1), $C_{17}H_{33}$ (for Crill 4)
$R_2$: H, or a fatty acyl group derived from poly condensed ricinoleic acid
$n$: the degree of polymerization of glycerol
$R_3$: Mono- or di-glyceride
$R_4$: H, or mono- or di-glyceride

**Figure 1.** The chemical structure of the emulsifiers used in the study (Crill™ 1 (sorbitan monolaurate), Crill™ 4 (sorbitan monooleate), PGPR 4150 (polyglycerol polyricinoleate), and AMP 4455 (ammonium phosphatide).

### 2.2. Casein Hydrolysate Production

First of all, casein was precipitated from skimmed milk by acidifying based on the method described in Sarode et al., (2016) with some modifications [40]. Skimmed milk was pasteurized by heating at 75 °C for 30 s in a heated, circulating water bath (IKA ICC Basic Eco 8). After pasteurization, milk was quickly cooled to 40 °C and kept at this temperature for 40 min. Then, milk was cooled to the precipitation temperature (35 °C). At this condition, 0.1 M acetic acid solution was slowly added with very gentle mixing until milk pH reached 4.6 and casein was precipitated by acidifying. Afterward, the mixture was heated to 50 °C and kept at this temperature for 15 min and whey was removed. Then, distilled water was added to the curd at 50 °C, and the curd was washed for 15 min with moderate stirring in a magnetic stirrer (IKA C-MAG HS 10). The washing water was removed using cheesecloth and the same washing process was repeated at 40 °C and 35 °C with distilled water. After each washing process, the excess water was removed from the casein curd using cheesecloth. The cheesecloth was hung up and left for 20 min. Finally, the cheesecloth was squeezed by hand for draining the whey, and acid casein was obtained.

The acid casein should be converted into a stable, homogenous fluid to provide effective hydrolysis. For this purpose, it was considered to bring the pH of acid casein to neutral pH with a phosphate buffer (0.1 M salt-free buffer containing $NaH_2PO_4$ and $Na_2HPO_4$ adjusted to pH 8.0). The incubations were carried out at a suitable temperature for enzyme activity (45 °C). Moreover, it was predicted that the hydrolysis process might continue for up to 72 h, and the highest acid casein concentration that would not lose its homogeneous dispersion structure, collapse, and/or adhere to the container walls in a shaking incubator for 72 h at 45 °C was determined during preliminary experiments. Accordingly, acid casein was crumbled by mixing at 7000 rpm for 60 s in a Thermomix TM31 (Vorwerk, Wuppertal, Germany) and phosphate buffer (containing 2% trisodium citrate, 1% trisodium phosphate salts, and 0.2% potassium sorbate on casein basis) was added to obtain a mixture with 1:3 acid casein: phosphate buffer ratio. Then, the mixture was heated to 80 °C with 4000 rpm mixing and the mixing process was continued for 15 min at 80 °C. Later on, the casein dispersion was rapidly cooled and 1.25% Alcalase was added. Samples were incubated at 45 °C for 8 h. At the end of the incubation, enzyme activity was terminated by a heat treatment at 90 °C for 15 min. Afterward, the hydrolysates were centrifuged at 8000 rpm for 10 min at room conditions and the supernatants were collected. Finally, the

collected supernatant was diluted 1:1 with phosphate buffer and used as dispersed water phase in the emulsions. The pH of the diluted casein hydrolysate was 7.2.

### 2.3. Preparation of W/O Emulsions

An appropriate amount of sunflower oil and emulsifier were mixed with a magnetic stirrer at room temperature. Crill™ 1, Crill™ 4, polyglycerol polyricinoleate, and ammonium phosphatide were used as emulsifiers in the present study and their abbreviations were written as C1, C4, PGPR, and AMP, respectively. After dissolving the emulsifier, the hydrolysate was added to the mixture. Then the mixture was homogenized with a rotor-stator (Ultra-Turrax, T18, IKA, Königswinter, Germany) and the shear force was gradually increased as follows: 5000 rpm for 30 s; 10,000 rpm for 30 s; and 15,000 rpm for 5 min.

Preliminary experiments were carried out to determine the highest dispersed phase ratio and the lowest emulsifier concentration to be considered as the restrictions for the study. After emulsion preparation, the pictures of the emulsions were taken (Figure 2). First, an emulsion containing 50% *w/w* distilled water as a dispersed phase and 50% *w/w* oil phase (2% *w/w* PGPR, and 48% *w/w* sunflower oil) was prepared. A proper emulsion was obtained (Figure 2a). After that, the emulsion was produced by using casein hydrolysate instead of distilled water as the dispersed phase with the same production technique. In this case, W/O emulsion could not form (Figure 2b). The reason for this can be related to the surface activity of the peptides and/or variation in the osmotic pressure due to the buffer solution in the hydrolysate solution. It was reported in the literature that the interfacial activity of peptides might interfere with PGPR at the W/O interface, leading again to destabilized emulsions [27]. In order to understand the effect of the buffer, the emulsion was prepared using the buffer solution as the dispersed phase (Figure 2c). Although emulsion formation seemed to be improved, proper W/O emulsion could also not be created. After that, the internal phase ratio was reduced to 40% (Figure 2d) and a better emulsion formation seemed to occur, whereas a stable W/O emulsion could not be obtained similar to the emulsion in Figure 2c. Finally, the emulsifier ratio was increased to 3% and a stable emulsion was formed (Figure 2e). Therefore, the lowest emulsifier ratio and the dispersed phase ratio to be used in experiments were decided as 3% and 40%, respectively.

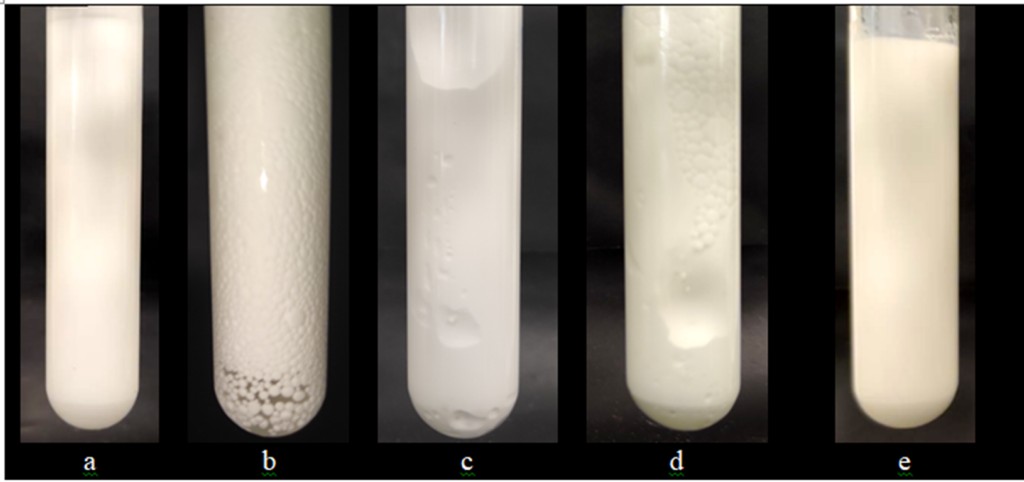

**Figure 2.** The emulsions containing (**a**) 50% *w/w* dispersed phase (distilled water) and 2% *w/w* PGPR, (**b**) 50% *w/w* dispersed phase (casein hydrolysate) and 2% *w/w* PGPR, (**c**) 50% *w/w* dispersed phase (buffer solution) and 2% *w/w* PGPR, (**d**) 40% *w/w* dispersed phase (casein hydrolysate) and 2% *w/w* PGPR, and (**e**) 40% *w/w* dispersed phase (casein hydrolysate) and 3% *w/w* PGPR.

Emulsions were also produced with other emulsifiers (Crill™ 1, Crill™ 4, and AMP) at the same conditions with the emulsion showed in Figure 2e. It has been observed that the emulsions produced with AMP present relatively high viscosity values. Therefore, the

emulsifier concentrations were reduced for emulsion prepared with AMP. After preliminary studies, the emulsion formulations used in the present study, and their codes are given in Table 1. A total of 36 emulsions were produced with 3 different dispersed phase ratios (ΦW), and 3 emulsifier concentrations (ΦE) for 4 emulsifiers (Table 1). Each production was duplicated and samples were analyzed immediately after the emulsion preparations.

**Table 1.** Compositions of W/O emulsions used in the present study and their sample codes *.

|  | ΦE: 3% *w/w* (for AMP 1% *w/w*) | ΦE: 5% *w/w* (for AMP 2% *w/w*) | ΦE: 7% *w/w* (for AMP 3% *w/w*) |
|---|---|---|---|
| ΦW: 10% *w/w* | 1 | 4 | 7 |
| ΦW: 25% *w/w* | 2 | 5 | 8 |
| ΦW: 40% *w/w* | 3 | 6 | 9 |

* ΦE: emulsifier concentration. ΦW: dispersed phase ratio.

### 2.4. Determination of Flow Behaviors

Flow behaviors of the emulsions was determined by viscometer (DV-II+ Pro Viscometer, Brookfield Engineering, Middleborough, MA, USA) coupled with a small sample adapter (SSA-13RD, Brookfield Engineering, Middleborough, MA, USA). The measurements were carried out at 35 °C and the temperatures of the samples were adjusted by a water circulation system (ICC Basic Eco 8, IKA, Staufen, Germany). Samples were analyzed with varying shear rates (in the range of 6.6–52.8 s$^{-1}$) and in addition to viscosity values, the flow behavior parameters (dimensionless flow behavior indices (n) and the consistency coefficients (K)) were calculated with the power-law model.

$$\tau = K(\dot{\gamma})^{n} \tag{1}$$

Here, $\tau$ is the shear stress, $\dot{\gamma}$ is the shear rate, *K* is the consistency coefficient, and *n* is the dimensionless flow behavior index.

### 2.5. Determination of Droplet Size Distribution by Microscopy-Assisted Digital Image Analysis

Droplet size distributions of emulsion samples were determined by a microscopy-assisted digital image analysis technique using Trainable Weka Segmentation as explained in detail by Salum et al., (2022) [41]. Firstly, micrographs were obtained by a compound microscope (M83EZ, OMAX Microscopes, Kent, WA, USA) combined with a 5-megapixel CMOS camera (A3550U, OMAX Microscopes, Kent, WA, USA). Briefly, 5 µL of the emulsion was diluted with 50 µL sunflower oil, dropped onto the microscope slide, and carefully covered with a coverslip. Immersion oil was dripped onto the coverslip and analysis was carried out with 100x/1.25 oil and a 160/0.17 objective lens. At least 120 photographs were taken from 5 separate microscope slides for each sample. Image analysis was performed using ImageJ/Fiji (ver.1.53c.) software. Trainable Weka Segmentation (TWS) (ver.3.2.35) plugin was used for the segmentation of emulsion droplets from the background. For this purpose, TWS was trained and a classifier was established. For the training, a total of 9 images were used which were selected from micrographs of emulsion prepared with PGPR, Crill™ 1, and Crill™ 4. After that, segmentations of the micrographs of the emulsions were performed using the trained TWS classifier. The images obtained as a result of TWS segmentation were turned into a single stack and this stack was converted to 8 bits and then transformed into the binary format. Afterward, the "Fill holes" and the "Open" commands were applied. Pixels corresponding to 10 µm were determined on the micrograph of the calibration slide and it was used as a scale for the particle size analyses. Eventually, the particle sizes of these images were calculated with a roundness value below 0.85. Over 8000 droplets were analyzed for each sample. By using the droplet size data, D(90), D [3, 2], and D [4, 3] values were calculated, which represent the equivalent volume diameters at 90%, and the area- and volume-weighted mean diameters, respectively.

### 2.6. Determination of Electrical Conductivity

Electrical conductivity values of the emulsions were determined with an electrical conductivity probe of pH/mV/EC/TDS/NaCl/Temp Bench Meter (MW 180Max, Milwaukee Instruments, Rocky Mount, NC, USA).

### 2.7. Monitoring the Emulsion Stability

The emulsion stability was observed visually by storing the emulsion samples at room temperature. For this purpose, emulsions were transferred to flat-bottom glass tubes and kept motionless. The emulsions were photographed for 5 days and the changes in their structure were evaluated. The emulsion stability was determined according to the occurrence of creaming or phase separation. Photographing was carried out in a box. The inner surface of the box was covered with a black cloth and the entrance of light from the outside into the box was prevented. A 7 W led light (daylight) was placed behind the test tubes to enhance the contrast during photographing.

### 2.8. Statistical Analysis

Experimental data were analyzed by performing ANOVA and Duncan post-hoc tests, and the statistical significance was $p < 0.05$. Moreover, principle component analysis (PCA) was performed to graphically show the relationship between the emulsions and their properties. SPSS statistical package program (SPSS ver. 22.0 for Windows, SPSS Inc., Chicago, IL, USA) and XLSTAT (Addinsoft, New York, NY, USA) were used in performing these statistical analyses.

The appropriate emulsion formulation according to the viscosity and droplet size values of the samples was determined using a multi-criteria decision analysis method called TOPSIS (the technique for order of preference by similarity to ideal solution). In this method, the chosen alternative should have the longest geometric distance from the negative ideal solution and the shortest geometric distance from the positive ideal solution [42]. While determining these positive and negative ideal solutions, first a matrix is constructed with the experimental results, and then the solution matrix is normalized and weighted. Briefly, the experimental results determine the boundary conditions for the ideal solutions [43]. The result of the TOPSIS was presented by the term called relative closeness (C) to the positive ideal solution. This value is between 0 and 1, and it is desirable to be close to 1.

## 3. Results and Discussion

### 3.1. Electrical Conductivity

Electrical conductivity measurement is a very useful method in evaluating the formation of W/O emulsions. While the electrical conductivity of the oil, which is the continuous/outer phase, is negligible, the dispersed/inner water phase (casein hydrolysate in the present study) shows a significant electrical conductivity. In this study, the electrical conductivity value measured for sunflower oil was $0.03 \pm 0.00$ µS/cm, and the same parameter for the casein hydrolysate prepared in buffer solution was detected as $13085 \pm 135$ µS/cm. The electrical conductivity of the emulsion is expected to be dominated by the continuous phase of the emulsion. Therefore, a negligible electrical conductivity value should be observed in an appropriate W/O emulsion and if a distinct electrical conductivity is observed in a sample, it means that the emulsion was not appropriately formed or lost its stability. In other words, a negligible electrical conductivity value indicates that the hydrophilic bioactive material is effectively encapsulated. In this context, the electrical conductivity values of the emulsions prepared in this study were checked immediately after the emulsion preparation. Among 36 samples, 31 samples did not have any electrical conductivity, only C1-3, C4-3, AMP-3, AMP-6, and AMP-9 had $2044 \pm 151$, $10400 \pm 593$, $1764 \pm 151$, $1675 \pm 155$, and $1228 \pm 83$ µS/cm, respectively. Moreover, rapid phase separation was also observed in these samples. Therefore, these samples were not analyzed in the continuation of the study.

### 3.2. Flow Behaviors

The flow behavior of emulsions is an important indicator of emulsion stability and properties. In this study, viscosity, flow behavior index, and consistency constant values were measured (Table 2), and higher dispersed phase ratios generally resulted in increased viscosity for W/O emulsions. In higher-concentration systems, the droplets begin to interact with each other through a combination of hydrodynamic and colloidal interactions [31].

**Table 2.** Flow behaviors of W/O emulsions produced using different emulsifiers with various emulsifier concentrations and dispersed phase ratios *.

| Sample | n (-) | K (Pa.s $^n$) | Viscosity (cP) |
|---|---|---|---|
| Crill 1-1 | 098 ± 0.00 [b,y] | 1.14 ± 0.00 [b,w] | 80.8 ±0.73 [b,x] |
| Crill 1-2 | 0.90 ± 0.00 [a,x] | 1.37 ± 0.04 [a,w] | 69.5 ± 0.41 [a,w] |
| Crill 1-4 | 0.95 ± 0.00 [a,m,x] | 1.18 ± 0.00 [c,k,w] | 74.5 ± 0.03 [a,w] |
| Crill 1-5 | 0.85 ± 0.03 [a,l,x] | 1.55 ± 0.17 [a,k,w] | 96.6 ± 2.57 [c,x] |
| Crill 1-6 | 0.63 ± 0.00 [k,w] | 2.66 ± 0.07 [l,w] | Non-Newtonian |
| Crill 1-7 | 0.95 ± 0.00 [a,l,y] | 1.10 ± 0.01 [a,k,w] | 92.7 ± 0.05 [c,x] |
| Crill 1-8 | 0.78 ± 0.07 [a,kl,x] | 1.95 ± 0.44 [a,kl,w] | 89.4 ± 0.49 [b,w] |
| Crill 1-9 | 0.70 ± 0.00 [k,w] | 2.66 ± 0.07 [l,x] | Non-Newtonian |
| Crill 4-1 | 0.96 ± 0.00 [b,xy] | 1.30 ± 0.02 [b,w] | 65.7 ± 0.60 [a,w] |
| Crill 4-2 | 0.86 ± 0.00 [a,x] | 1.37 ± 0.04 [a,w] | 83.5 ± 2.82 [a,y] |
| Crill 4-4 | 0.97 ± 0.01 [b,m,xy] | 1.15 ± 0.03 [a,k,w] | 77.6 ± 0.23 [b,x] |
| Crill 4-5 | 0.86 ± 0.00 [a,l,x] | 1.35 ± 0.01 [a,l,w] | 80.8 ± 0.80 [a,w] |
| Crill 4-6 | 0.63 ± 0.01 [k,w] | 2.62 ± 0.04 [m,w] | Non-Newtonian |
| Crill 4-7 | 0.88 ± 0.00 [a,l,x] | 1.20 ± 0.03 [ab,k,w] | 77.0 ± 0.62 [b,w] |
| Crill 4-8 | 0.81 ± 0.02 [a,kl,x] | 1.67 ± 0.17 [a,k,w] | 85.0 ± 1.09 [a,w] |
| Crill 4-9 | 0.71 ± 0.04 [k,w] | 1.76 ± 0.26 [k,w] | Non-Newtonian |
| PGPR-1 | 0.93 ± 0.00 [a,k,x] | 1.04 ± 0.03 [a,k,w] | 83.9 ± 0.23 [a,k,y] |
| PGPR-2 | 1.04 ± 0.03 [a,l,y] | 1.19 ± 0.11 [a,k,w] | 132.5 ± 0.39 [a,l,y] |
| PGPR-3 | 0.98 ± 0.00 [a,kl] | 2.79 ± 0.03 [a,l] | 260.2 ± 0.95 [a,m] |
| PGPR-4 | 0.99 ± 0.01 [b,k,y] | 1.04 ± 0.04 [a,k,w] | 93.4 ± 0.33 [b,k,y] |
| PGPR-5 | 1.01 ± 0.01 [a,k,y] | 1.36 ± 0.02 [a,l,w] | 139.8 ± 0.44 [b,l,y] |
| PGPR-6 | 0.98 ± 0.00 [b,k,x] | 2.83 ± 0.01 [a,m,w] | 271.9 ± 0.58 [b,m] |
| PGPR-7 | 1.01 ± 0.00 [b,l,z] | 1.02 ± 0.00 [a,k,w] | 100.1 ± 1.26 [c,k,y] |
| PGPR-8 | 1.01 ± 0.01 [a,l,y] | 1.40 ± 0.00 [a,l,w] | 146.0 ± 1.48 [c,l,x] |
| PGPR-9 | 0.99 ± 0.00 [b,k,x] | 2.84 ± 0.01 [a,m,x] | 274.7 ± 1.56 [b,m] |
| AMP-1 | 0.43 ± 0.01 [a,w] | 11.4 ± 0.38 [a,x] | Non-Newtonian |
| AMP-2 | 0.28 ± 0.01 [b,w] | 95.3 ± 4.08 [a,x] | Non-Newtonian |
| AMP-4 | 0.45 ± 0.00 [ab,w] | 10.1 ± 0.33 [a,x] | Non-Newtonian |
| AMP-5 | 0.21 ± 0.01 [a,w] | 145.2 ± 9.25 [b,x] | Non-Newtonian |
| AMP-7 | 0.48 ± 0.01 [b,w] | 9.5 ± 0.78 [a,x] | Non-Newtonian |
| AMP-8 | 0.31 ± 0.00 [b,w] | 133.1 ± 4.62 [b,x] | Non-Newtonian |

* Abbreviations are n, flow behavior index; K, consistency coefficient. Values are mean ± standard deviation of the analysis results and the same superscript letters ([a–c] for emulsifier concentration; [k–m] for dispersed phase ratio; [w–z] emulsifier type) indicate no significant difference between the samples produced at different conditions ($p > 0.05$).

In addition, the results show that the type of emulsifier had a significant effect on the viscosity and the flow behavior of the emulsions. Emulsions produced with PGPR behaved as Newtonian-type fluid and their viscosity increased with dispersed phase ratio and emulsifier concentration. Similarly, Jo et al., (2019) determined Newtonian flow properties in collagen peptide hydrolysate-loaded primary emulsions prepared using PGPR and also they noted that the viscosity increased with increasing internal phase [24]. In emulsions prepared with sorbitan esters (both Crill™ 1 and Crill™ 4), the dispersed phase ratio dominated the flow behavior of emulsions. It was found that samples 6 and 9, with the highest internal phase ratio (40%), had significantly higher K values and showed non-

Newtonian flow behaviors. It is reported that the Newtonian character of both W/O and O/W emulsions changes and the emulsions exhibit non-Newtonian behaviors with the increase in the dispersed phase ratio, especially before the phase inversion [44–46]. Samples prepared with AMP differed from the others in terms of flow behavior and consistency. All formulations produced with AMP had significantly higher K values compared to other emulsifiers even though they were used in lower ratios. Moreover, emulsions prepared with AMP showed non-Newtonian flow behavior regardless of the dispersed phase ratio. Shear thinning behavior of AMP-stabilized W/O were reported by Rivas et al., (2016), and also, higher viscosity values were obtained in AMP-stabilized W/O than in PGPR-stabilized ones [47]. According to the micrographs of emulsions prepared with AMP, droplets were not completely dispersed and seemed to be aggregated (Figure 3). Rivas et al., (2016) and Balcaen et al., (2017) observed the same trends with light microscopy [47,48]. Emulsions prepared using AMP were highly aggregated systems, while emulsions stabilized with PGPR had very small, individual droplets [47,48]. The emulsions prepared in this study were designed as the inner phase of double emulsions and were aimed to be of low viscosity. Since the viscosity of the emulsions produced with AMP was very high, the emulsions produced with this emulsifier were not included in the later parts of the study.

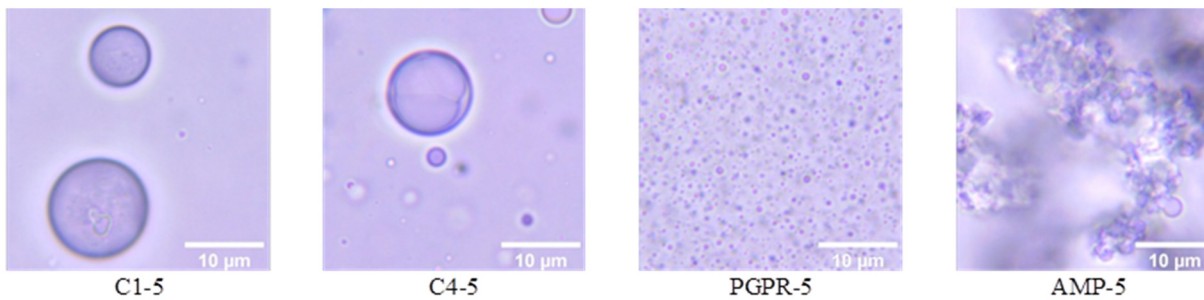

**Figure 3.** Micrographs of W/O emulsion produced with constant 25% dispersed phase ratio and medium-level emulsifier concentrations (5% for Crill™ 1, Crill™ 4, and PGPR and 2% for AMP).

### 3.3. Emulsion Stability

The gravimetric approach was used to determine the emulsion stability. W/O emulsions stabilized with Crill™ 1, Crill™ 4, and PGPR was examined through photographs taken during the 5-day storage period (Figure 4). It is observed that Crill™-1- and Crill™-4-stabilized emulsions had district phase separation after 24 h. However, no visual phase separation was detected in the PGPR-stabilized samples even after 5 days. PGPR produced significantly stable casein-hydrolysate-loaded W/O emulsions compared to others in all formulations. The highly effective applications of PGPR in stabilizing W/O emulsions have been reported by several authors in the literature [15,47,49]. In one of these studies, PGPR was used to produce peptide-loaded emulsions [15]. Ying et al., (2021) determined that the PGPR-stabilized soy-peptide-loaded W/O emulsion presented better stability than the Span-60- and lecithin-stabilized ones [15].

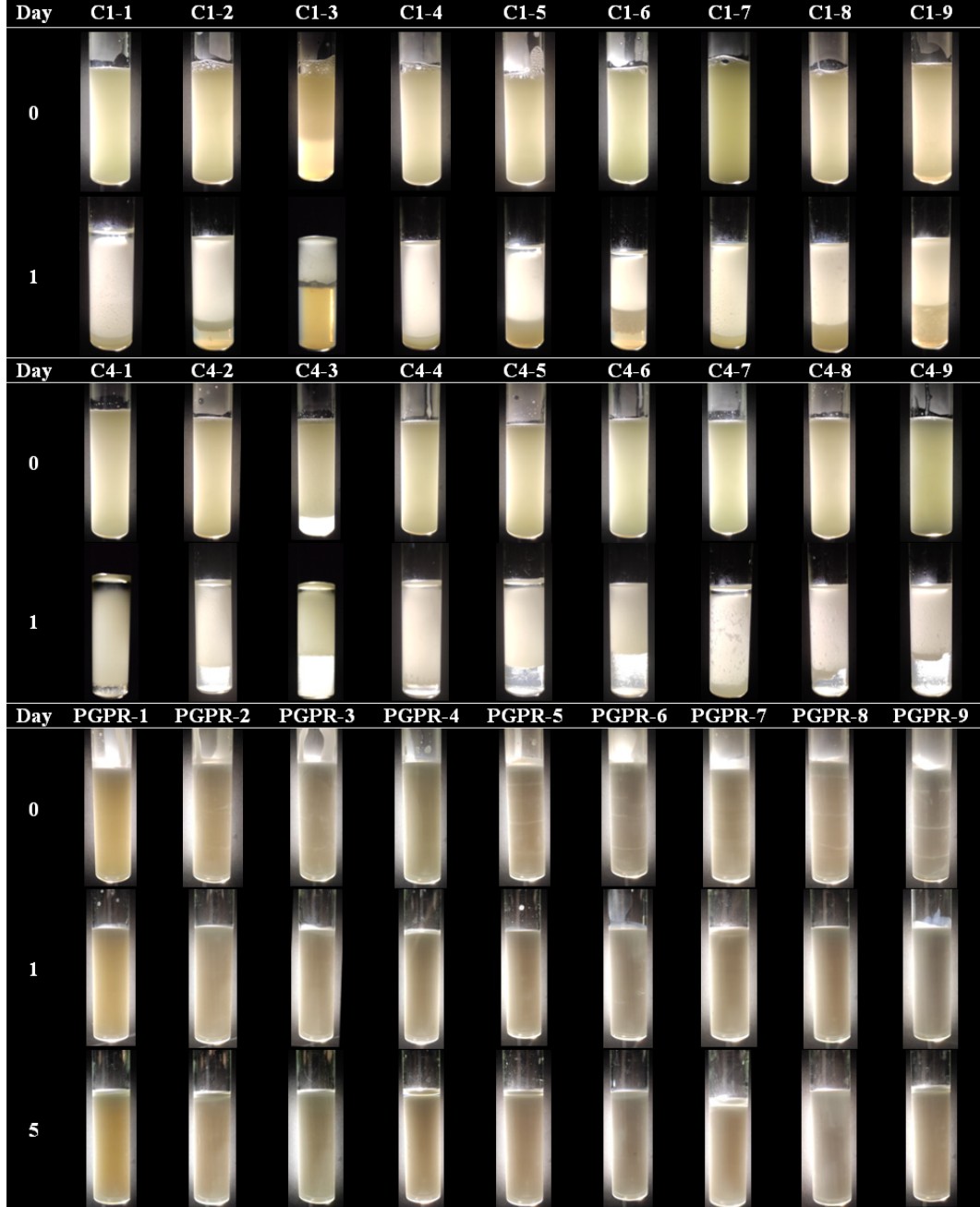

**Figure 4.** The gravimetric stability of W/O emulsions stabilized with Crill™ 1, Crill™ 4, and PGPR.

### 3.4. Droplet Size Distributions

The droplet size and size distribution in emulsions are important parameters on stability, as an increase in droplet size can lead to destabilization of the emulsions by flocculation, coalescence, or Oswalt ripening [31]. The droplet size distributions of emulsions can differ for various reasons, such as emulsification conditions, type and amount of emulsifier, and the ratio of dispersed and continuous phases [24]. As seen in Figure 5, the droplet size of the Crill™ 1 and Crill™ 4 samples were remarkably varied from the droplet sizes of the PGPR samples. This was also observed in micrographs (Figure 3). According to the comparison of the emulsifiers with the same formulations, D(90) values of Crill™-1- and Crill™-4-stabilized emulsions were found 9.5 to 20.8 and 6.7 to 21.4 times higher than PGPR stabilized emulsions, respectively. These results were in harmony with the emulsion

stability results (Figure 4). While D(90) values in all PGPR-stabilized emulsions were below 2 μm, D [4, 3], and D [3, 2] values were less than 1 μm.

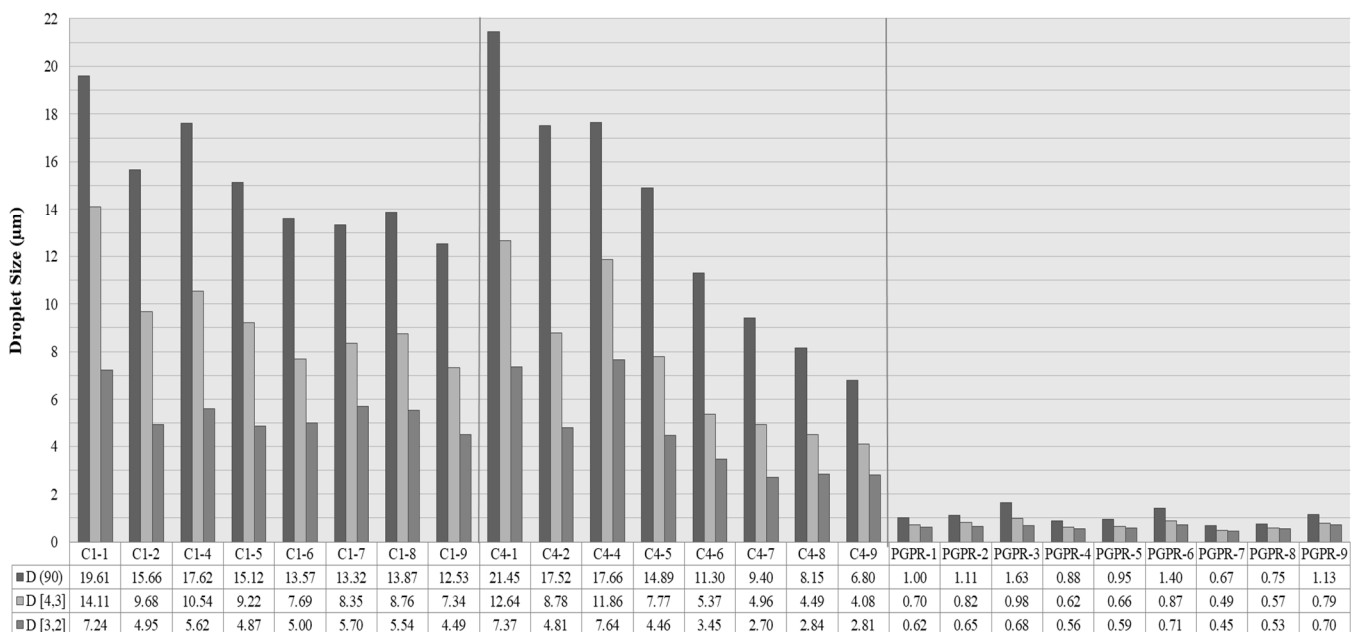

**Figure 5.** The droplet size parameters for W/O emulsions were prepared with different emulsifiers and formulations.

PCA was performed and a PCA bi-plot was drawn to show the closeness of the samples with droplet size parameters and consistency coefficient (Figure 6). The calculated principal components (F1 and F2) explained a very important part of the variations, e.g., 99.1% and 75.9% of the variations that could be represented by the component (F1) are shown on the x-axis alone. According to PCA analysis, all samples produced with PGPR were located away from the droplet size parameters along the x-axis. However, PGPR samples containing 40% internal phase were positioned at the positive F2; likewise, the consistency coefficient and other PGPR stabilized samples were positioned at the negative side of the y-axis. In addition, a negative correlation concerning F1 was found between the consistency coefficient and the droplet size parameters. This negative correlation may be due to several possible reasons. As the droplet size decreases, the average separation distance between the droplets also decreases, resulting in an increase in hydrodynamic interaction and viscosity. Furthermore, the increases in viscosity upon droplet size reduction may be due in part to an increase in the effective dispersed phase concentration. In other words, as the droplet size decreases, the thickness of the adsorbed emulsifier layer relative to the droplet size becomes more important. Finally, it is known that the polydispersity generally decreases with the decrease in droplet size and influences the flow behaviors [50].

**Biplot (axes F1 and F2: 99.10 %)**

**Figure 6.** PCA bi-plot diagram for droplet size parameters and consistency coefficient for W/O emulsions prepared with different emulsifiers and formulations.

*3.5. Determination of Appropriate Emulsion Formulation*

In the present study, it was determined that all samples prepared with PGPR had low droplet sizes with high emulsion stability. Rivas et al., (2016) also highlighted that PGPR emulsions have mono-modal droplet size distribution with smaller droplets without networking tendency, and these emulsions showed Newtonian flow behavior with much higher stability to phase separation. As a result of these features, using PGPR in the formulation of the primary emulsion during the production of the double emulsion was suggested [47]. Considering that, PGPR was selected as the most suitable emulsifier for the preparation of primary emulsion.

Although the results in this study were evident and showed that PGPR would be appropriate for the use in the formation of a primary emulsion of the double emulsion, it is not clear in which formulation it would be preferred. For this purpose, TOPSIS analysis was performed to determine the most appropriate PGPR-stabilized emulsion prepared using low emulsifier concentrations, high dispersed phase ratio at low emulsion viscosity, and droplet size. In TOPSIS, dispersed phase ratio, emulsifier concentration, viscosity, and droplet size (D [4, 3]) parameters were used as responses, and their weights were decided as 20%, 20%, 30%, and 30%, respectively. The emulsion with the highest relative closeness value calculated by TOPSIS was selected. It was seen that three emulsion formulations (PGPR-1, PGPR-2, and PGPR-5) gave very close relative closeness values above 0.6, and PGPR-5 (25% dispersed phase ratio and 5% emulsifier concentration) had the highest value as 0.621 (see Table 3).

**Table 3.** Technique for order of preference by similarity to ideal solution (TOPSIS) similarity values for the emulsions prepared by PGPR 4150.

| Code | Dispersed Phase (%) | PGPR (%) | Viscosity (cP) | D [4, 3] (µm) | Relative Closeness (-) |
|---|---|---|---|---|---|
| PGPR-1 | 10 | 3 | 83.9 | 0.704 | 0.611 |
| PGPR-2 | 25 | 3 | 132.5 | 0.822 | 0.616 |
| PGPR-3 | 40 | 3 | 260.2 | 0.976 | 0.430 |
| PGPR-4 | 10 | 5 | 93.4 | 0.618 | 0.591 |
| PGPR-5 | 25 | 5 | 139.8 | 0.662 | 0.621 |
| PGPR-6 | 40 | 5 | 271.9 | 0.873 | 0.396 |
| PGPR-7 | 10 | 7 | 100.1 | 0.488 | 0.568 |
| PGPR-8 | 25 | 7 | 146.0 | 0.575 | 0.572 |
| PGPR-9 | 40 | 7 | 274.7 | 0.795 | 0.381 |

## 4. Conclusions

The results of the present study showed that the selection of emulsifier type is an important cornerstone for obtaining more stable primary emulsions, which is critical for the stability of double emulsions. Thus, the results of experimental studies clearly showed that PGPR is the most suitable emulsifier type among other types due to its ability to form emulsions with relatively low viscosity and significantly high stability. The most appropriate formulation with PGPR-based primary emulsion was a 25% of dispersed phase ratio and 5% of emulsifier concentration in the dispersed (oil) phase. This emulsion formulation resulted in a viscosity of 139.8 cP and a D [4, 3] value of 0.662 µm. In summary, the detailed characterization of emulsions provides the true selection of emulsifier type in terms of more stable primary emulsion and accordingly high-quality double emulsions.

**Author Contributions:** Conceptualization, Z.E.; methodology, P.S. and Z.E.; validation, P.S. and Ç.U.; formal analysis, P.S., Ç.U. and O.G.; investigation, P.S. and Z.E.; resources, O.G., L.Y.A. and Z.E.; data curation, P.S. and Ç.U.; writing—original draft preparation, P.S. and Z.E.; writing—review and editing, O.G. and L.Y.A.; visualization, P.S.; supervision, L.Y.A.; project administration, Z.E.; funding acquisition, L.Y.A. and Z.E. All authors have read and agreed to the published version of the manuscript.

**Funding:** This work was supported by the Scientific and Technological Research Council of Turkey (TUBITAK) (Project No. 120O763).

**Institutional Review Board Statement:** Not applicable.

**Informed Consent Statement:** Not applicable.

**Data Availability Statement:** Not applicable.

**Acknowledgments:** The authors are also grateful to Novozymes for the enzymes provided and Palsgaard and Croda for the emulsifiers supplied.

**Conflicts of Interest:** The authors declare no conflict of interest.

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
