# Peer review of "Casein-Hydrolysate-Loaded W/O Emulsion Preparation as the Primary Emulsion of Double Emulsions: Effects of Varied Phase Fractions, Emulsifier Types, and Concentrations"

_colloids, doi:10.3390/colloids7010001_

Round 1
Reviewer 1 Report
The manuscript presented an interesting study on optimizing the formulations for a stable primary emulsion. The manuscript can be published if the following comments are properly addressed.
1. It is good to show the chemical structure of the emulsifiers for readers to better understand their interfacial activity.
2. What are the CMCs of the emulsifiers? The effective concentration of the emulsifiers in stabilizing the emulsions is supposed to be related to the CMC. To compare the effectiveness of the emulsifiers, it is important to have the numbers of their CMCs.
3.
4. Figure 1. Please indicate when the pictures were taken after homogenizing the mixture (emulsion preparation).
5. Line 263. “the same value” should be “the same parameter (electrical conductivity)”. The electrical conductivities of sunflower oil and casein hydrolysate differ dramatically. Their values are not the same.
6. In TOPSIS method, what are the negative and positive ideal solutions? How are they selected and what are the physics behind them?
7. Table 3. The table caption should be “….emulsions prepared by PGPR 4150.”
8. Lines 411-412. “… that PGPR is the most suitable emulsifier type among other types due to its relatively low viscosity and significantly high stability properties.” Please rephase the sentence to make it clear that “relatively low viscosity and significantly high stability properties” is for the emulsion, not the emulsifier.
9. Line 413. “…was 5% of emulsifier concentration…” Please make it clear that the emulsifier concentration is with respect to the oil phase, not to the emulsion.
10. Line 415. “0.662 2 μm” should be 0.662 μm.
11. Lines 417-418 in conclusion are not supported in the manuscript (not discussed).
Author Response
On behalf of my co-authors, I would like to take this opportunity to register my sincere appreciation to you as the reviewer for the valuable comments, which helped us improve the quality of the paper.
We have tried to do our best based on your valuable comments. We hope that our answers to the queries presented in the text (as yellow highlighted) and the response sheet document attached will be sufficient for the acceptance of the paper.
With my kindest regards,
Zafer Erbay, Ph.D.
The Corresponding Author

Reviewer 2 Report
Comments
This manuscript titled “Peptide-loaded W/O emulsion preparation as the primary emulsion of double emulsions: Effects of varied phase fractions, emulsifier types, and concentrations”(ID No. colloids-2069947) explores the effect of different emulsifiers (different emulsifiers (Crill™ 1, Crill™ 4, AMP, and PGPR) on the fabrication of peptide-loaded stable primary emulsion.However, the background and significance of this research are not sufficient. Some results could be further discussed to make manuscript more complete.The following are major points that need to be addressed.
1. For the part Introduction, it is too long and the information of W/O/W emulsions was too much to conceal the importance of this research. Actually, this work mainly explored the primary emulsion. Please revise it and make the highlights clear.
2. Line 92 and Line 94, “peptide” and “caseinhydrolysates” were not coincident, and after reading this manuscript, casein hydrolysates might be proper.
3. Line 111, casein was extracted from milk, so why did authors use commercial casein? Or did authors consider the difference between them?
4. Line 141, how to ensure the complete inactivation of enzyme activity?
5. The part of “2.3 Preparation of W/O emulsions”, how about the pH values of these emulsions?
6. Line 153, the written mistake of the sentence “10.000 for 30 sec, and 15.000 for 5 min”, thus 10.000 revised into 10,000 and 15.000 revised into 15,000, please.
7. Line 157-158, Figure 1a only contains distilled water but can obtain stable emulsion, while Figure 1b is obviously unstable. Protein hydrolysate should also have a certain interfacial tension, but why is it less stable than pure water?
8. Line 195, what is the range of shear rates?
9. Line 201, the part “size distribution”, Why did not use instrumentation to measure it? How reliable are the results of the microscope?
10. The part “2.7 Monitoring the Emulsion Stability”, What’s the criteria for emulsion stability? Is creaming height? This should be fully stated in the methods.
11. For results, it would be better to make some unimportant results as the supplement files rather than the manuscript.
12. After reading this manuscript, the peptide was not well presented. What is the purpose of it? If authors treat it as activity compounds, they should exhibit the encapsulation efficiency or loading efficiency?
Author Response

(The authors gave the same response as above.)

Round 2
Reviewer 2 Report
No more comments.